

# Rivaroxaban, a direct inhibitor of coagulation factor Xa, attenuates adverse cardiac remodeling in rats by regulating the PAR-2 and TGF-β1 signaling pathways

Qian Zhang, Zhongfan Zhang, Weiwei Chen, Haikuo Zheng, Daoyuan Si and Wenqi Zhang

Department of Cardiology, China-Japan Union Hospital of Jilin University, Changchun, Jilin, China

## ABSTRACT

**Background**. Factor Xa (FXa) not only plays an active role in the coagulation cascade but also exerts non-hemostatic signaling through the protease-activated receptors (PARs). This study aimed to investigate whether the FXa inhibitor, Rivaroxaban (RIV), attenuates adverse cardiac remodeling in rats with myocardial infarction (MI) and to identify the underlying molecular mechanisms it uses.

**Methods**. An MI model was induced in eight-week-old, male Wistar rats, by permanent ligation of the left anterior descending coronary artery. MI rats were randomly assigned to receive RIV or protease-activated receptors 2-antagonist (PAR-2 antagonist, FSLLRY) treatment for four weeks. Histological staining, echocardiography and hemodynamics were used to assess the cardioprotective effects of RIV. Meanwhile, pharmacological approaches of agonist and inhibitor were used to observe the potential pathways in which RIV exerts antifibrotic effects in neonatal rat cardiac fibroblasts (CFs). In addition, real-time PCR and western blot analysis were performed to examine the associated signaling pathways.

**Results**. RIV presented favorable protection of left ventricular (LV) cardiac function in MI rats by significantly reducing myocardial infarct size, ameliorating myocardial pathological damage and improving left ventricular (LV) remodeling. Similar improvements in the PAR-2 antagonist FSLLRY and RIV groups suggested that RIV protects against cardiac dysfunction in MI rats by ameliorating PAR-2 activation. Furthermore, an *in vitro* model of fibrosis was then generated by applying angiotensin II (Ang II) to neonatal rat cardiac fibroblasts (CFs). Consistent with the findings of the animal experiments, RIV and FSLLRY inhibited the expression of fibrosis markers and suppressed the intracellular upregulation of transforming growth factor β1 (TGFβ1), as well as its downstream Smad2/3 phosphorylation effectors in Ang II-induced fibrosis, and PAR-2 agonist peptide (PAR-2 AP) reversed the inhibition effect of RIV.

**Conclusions**. Our findings demonstrate that RIV attenuates MI-induced cardiac remodeling and improves heart function, partly by inhibiting the activation of the PAR-2 and TGF-β1 signaling pathways.

Corresponding authors
Daoyuan Si, sidaoyuan@jlu.edu.cn
Wenqi Zhang, wenqi@jlu.edu.cn,
Zhangqianmed@163.com

# INTRODUCTION

Several landmark randomized clinical trials recently showed that rivaroxaban (RIV), a factor Xa (FXa) inhibitor, provides some clinical benefits that have not been observed with conventional anticoagulation regimens in patients with cardiovascular disease (*Branch et al., 2019*; *Eikelboom et al., 2017*; *Mega et al., 2012*). However, these benefits were not observed in previous studies on vitamin K antagonists, suggesting that RIV may play a non-haemostatic role, independent of its effect on the coagulation cascade. FXa is a key serine protease at the confluence of the endogenous and exogenous pathways of the coagulation cascade (*Ten Cate et al., 2021*). Growing evidence has demonstrated the important role of FXa in maladaptive cardiac fibrosis, hypertrophy and remodeling during tissue injury, beyond its effects on coagulation (*Kondo et al., 2018*; *Narita et al., 2021*). Adverse left ventricular (LV) remodeling after myocardial infarction (MI), accompanied by cardiomyocyte hypertrophy, cardiomyocyte death, inflammatory and fibrotic responses, which often culminate in heart failure (HF), remains a major source of late morbidity and mortality after MI (*Hill & Olson, 2008*; *Sutton, 2000*). Myocardial fibrosis usually manifests as a diffuse and disproportionate accumulation of type I and type III collagen fibers in the myocardial interstitium, forming a reparative fibrotic scar that may act as a natural protective mechanism against cardiac rupture (*Porter & Turner, 2009*). However, excessive interstitial fibrosis may lead to the progressive deterioration of cardiac function (*Prabhu & Frangogiannis, 2016*). Direct inhibition of FXa by RIV may be a promising cardioprotective strategy for the treatment of adverse ventricular remodeling.

Recent evidence shows that FXa modulates non-hemostatic cellular responses, largely through protease-activated receptors 1 and 2 (PAR-1 and PAR-2)-dependent mechanisms (*Papadaki & Tselepis, 2019*), which mediate the activation of canonical G protein pathways and are widely expressed in several cardiac cells, including cardiomyocytes and cardiac fibroblasts (*Antoniak, Sparkenbaugh & Pawlinski, 2014*). Previous studies have shown that the PAR-2 signaling pathway plays a role in the regulation of myocardial hypertrophy and myocardial ischemic remodeling (*Antoniak et al. 2010*)). Moreover, PAR-2 has been reported to initiate fibrotic signaling in chronic organ injury using a transactivation mechanism (*Chung et al., 2013*; *Knight et al., 2012*). For example, PAR-2 works synergistically with the TGF-β1 signaling pathway to promote renal injury and fibrosis (*Chung et al., 2013*). PAR-2 agonists enhance TGF-β1 and the expression of other profibrotic genes to exert a pro-fibrotic effect on hepatic stellate cells (*Knight et al., 2012*). Results of another study suggested that PAR-2 was a novel regulator of TGF-β signaling in pancreatic cancer (*Witte et al., 2016*). Based on these data, it is reasonable to speculate that the FXa inhibitor, RIV, may exert cardioprotective effects on the pathological development of myocardial fibrosis through PAR-2 and TGF-β1 signaling.

RIV has shown protective effects in a variety of mouse models. The study conducted by *Nakanishi et al. (2020)* found that RIV protected against cardiac dysfunction by inhibiting PAR-1, PAR-2 and proinflammatory cytokines, with effects observed after seven days in MI model mice. Another study conducted by *Bode et al. (2018)* showed that RIV did not reduce cardiac dysfunction in PAR-$2^{-/-}$ mice. An experimental model of intermittent hypoxia showed that RIV and a PAR-2 antagonist attenuate both atrial and ventricular remodeling by preventing oxidative stress and fibrosis (*Imano et al., 2018*). However, the detailed underlying mechanism of such cardioprotective effects in above study was not demonstrated. Build on the existing knowledge of the RIV, the present study extended previous observations of the cardioprotective effects of RIV. In this study, we aimed to investigate the effectiveness of RIV in reducing adverse cardiac remodeling and improving cardiac function in rats that 28 days after MI. Furthermore, we delved deeper into the mechanism of action of the RIV and make this knowledge more comprehensive.

## MATERIALS & METHODS

### Animal experiments

All animal experiments were approved by the Ethics Committee of the First Hospital of Jilin University (protocol code: 0755) and conducted in accordance with the Guide for the Care and Use of Laboratory Animals. Eight-week-old, male Wistar rats, weighing between 200–250 g, were purchased from the Vital River Laboratory Animal Technology Co., Ltd. (Beijing, China). The rats were housed, individually, in an environment controlled for ambient temperature (20–22 °C) and humidity (45–50%), with a 12-h light/12-h dark cycle. All rats had free access to food and water during the study. All rats were euthanized with 1% sodium pentobarbital (40 mg/kg, i.p.), sterilized, and buried at the end of the study. The MI model was established by left coronary artery anterior descending (LAD) branch ligation, as previously described by *Muthuramu et al. (2014)*. Rats in the sham operation group underwent the same operation without ligation. Rats were given RIV or FSLLRY immediately after recovery from anesthesia after LAD ligation or sham surgery. rats in the RIV group had 3 mg/kg/day of RIV intragastrically administered (MCE; BAY59-7939; *Guillou et al., 2020*), while rats in the FSLLRY group received an intraperitoneal injection with 10 ug/kg/day of PAR-2 peptide antagonist (FSLLRY, Tocris, R & D System Biotechne; *McLarty et al., 2011*) for four weeks. There were forty rats included in the study with ten rats in each group. Simple random sampling was done using RANDOM.ORG. Each rat was given a number before randomization. After the four-week experimental period, cardiac function was evaluated by echocardiography. Part of the left ventricle was used for subsequent histological evaluation, and part was used for western blotting, immunohistochemistry, and reverse transcription polymerase chain reaction (RT-PCR) (Fig. 1A).

### Echocardiography

To assess changes in cardiac structure and function in a noninvasive manner, echocardiography (GE VIVId-i) was performed four weeks post-MI. Rats remained anesthetized during the entire echocardiography procedure. Parameters of left ventricular
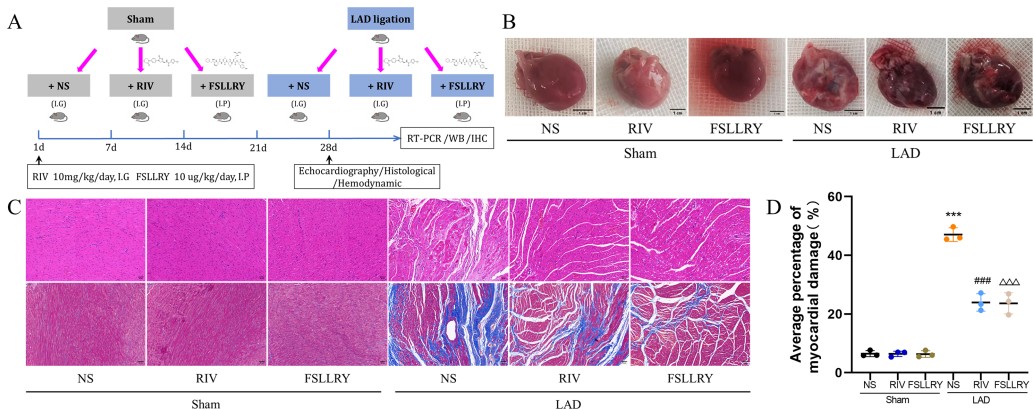

**Figure 1** **RIV and FSLLRY improves cardiac morphological changes in MI Rats.** (A) Experimental scheme whereby rats were subjected to LAD surgery or a sham procedure for four weeks. Eight-week-old rats were treated with RIV or FSLLRY until harvesting. (B) The overall appearance of the rat heart. T scale =1 cm. (C) Representative images of H&E and Masson's trichrome staining in hearts after four weeks, scale bar = 50 $\mu$m. (D) Quantitative analysis of collagen deposition in the left ventricle; $n = 3$; ***$P < 0.001$, Sham *vs* LAD; ###$P < 0.001$, LAD *vs* LAD+RIV; $\triangle\triangle\triangle P < 0.001$, LAD *vs* LAD+FSLLRY. IG, intra-gastrical; IP, intraperitoneal; WB, western blotting; IHC, immunohistochemical.

function were measured from M-mode images obtained from a short-axis view. The left ventricular ejection fraction (EF) and the percentage of fractional shortening (FS) were calculated from the measured dimensions. The echocardiographer was blinded to the treatment groups.

## Hemodynamic assessment

After echocardiography was completed, a hemodynamic assessment was performed. The skin of the anterior portion of the neck was incised, anterior muscles were separated, and the internal carotid artery was exposed. Then, the internal carotid artery was carefully punctured with a 23G syringe needle. An invasive hemodynamic catheter (FPI-LS-PT9; FISO Technology Inc., QC, Canada) was then placed into the lumen of the internal carotid artery along the vessel breach. All signals were analyzed offline using a signal conditioner and the data analyses were performed using LabScribe software. The following metrics were evaluated: left ventricular systolic pressure, and maximum and minimum left ventricular pressure development rate ($\pm$dP/dt).

## Histological examination

Heart tissues were fixed in 4% paraformaldehyde for 24 h, embedded in paraffin, cut into 4-um sections and subsequently stained with hematoxylin and eosin (H&E) and Masson's trichrome stain. After gradient ethanol dehydration, infiltration and sealing, the sections were observed and photographed using a microscope (OLYMPUS BX53) at 200x magnification. Three fields of view were randomly tested for each sample. The degree of fibrosis was assessed by quantifying the deposited collagen (blue areas on the section) relative to the total tissue area on Masson's trichrome-stained tissue sections.

Image analysis was performed using Image-Pro Plus 6.0 software (Media Cybernetics, Rockville, MD, USA).

## Western blotting

Proteins were isolated from snap-frozen cardiac tissue and cultured CFs using RIPA lysis buffer containing protease and phosphatase inhibitors. The protein concentration was determined using the BCA Protein Assay Kit (PC0020; Solarbio). After denaturation, the proteins were separated with SDS-page and transferred to PVDF membranes. After blocking with 5% BSA, the membranes were incubated overnight at 4 °C with specific antibodies against the following proteins: PAR-2, 1:1000, Bioss, bs-1178R; TGFβ1, 1:1000, Abcam, ab179695; α-SMA, 1:1000, Cell Signaling Technology, 19245; p-Smad2, 1:1000, Bioss, bs-3419R; Smad2, 1:1000, CST, 5339S; p-Smad3, 1:1000, abclonal, AP0727; Smad3, 1:1000, abclonal, A19115; Collagen I, 1:1000, Abcam, ab260043; Collagen III, 1:1000, Cell Signaling Technology, 66887; and GAPDH, 1:1000, Abcam, ab181602. HRP-conjugated IgG was used for the secondary antibodies, and an ECL Chemiluminescent Detection Kit (Sparkjade) was used to detect antibody–antigen complexes. Then, each nitrocellulose membrane was stripped with stripping buffer (1% SDS, 25 mM glycine [pH 2.0]) and then incubated with GAPDH primary antibody. The immunoblots were quantified using Image-Pro Plus 6.0 software (Media Cybernetics). Grey values of the duplicate data strips for western blot were calculated from image J, using the following process in the software: 1. Image-Type-8bit; 2. Process-Subtract Background: Light background. Rolling ball radius: 50.0 pixel; 3. Analyze-Set Measurements: area, min & max gray value, integrated density, mean gray value; 4. Analyze-set scale-distance in pixel: 0, Known distance: 0, Pixel aspect ratio: 1.0, Unit of length: pixel; 5. Edit-Invert-choice the target strip- analyze, and export as the data file Excel. Histograms were exported using Prism8.

## Immunohistochemical analysis (IHC)

The heart tissues were fixed in paraformaldehyde, embedded in paraffin, dewaxed and rehydrated, then blocked in 5% albumin from bovine serum (BSA; Solarbio) for 30 min at room temperature. The tissues were then incubated with primary antibody against PAR-2 (1:100), α-SMA (1:400) or TGF-β(1:1500) and diluted in 5% BSA overnight at 4 °C. Next day, Sections were incubated for 1 h with biotin-labeled goat anti-rabbit IgG (1:4000, bs-80295G-HRP), and then stained with DAB buffer for another 15 min at room temperature. Stained sections were observed and photographed using ScanScope CS (Leica Biosystems, Wetzlar, Germany). Finally, Stained sections were observed and photographed using microscope (Zeiss, Germany). The negative control group were incubated with 5% BSA without primary antibody overnight at 4 °C and the following procedures were same.

## qRT-PCR analysis

Total RNA was extracted from fresh heart tissues and CFs with Trizol reagent, and the RNA concentration was determined using spectrophotometry. Total RNA was reverse transcribed into cDNA according to the manufacturer's instructions. The quantitative RT-PCR analysis was performed on a real-time PCR system (ABI 7500 DX) using SYBR Green Master Mix (Sparkjade), the GAPDH gene was used to standardize mRNA expression,

and the expression was calculated by $2-\Delta\Delta CT$. Primers utilized in this research were: Collagen I: forward 5′- TGG TCT TGG AGG AAA CTT TGC -3′, reverse 5′- CTG TGT CCC TTC ATT CCGG -3′; Collagen III: forward 5′- ACC TGA AAT TCT GCC ACCCT -3′, reverse 5′- GCC TTG AAT TCT CCC TCA TTG -3′; α-SMA: forward 5′- AAA TGA CCC AGA TTA TGT TTG AGAC', reverse 5′- CAT CTC CAG AGT CCA GCA CAA TAC -3′; GAPDH: forward 5′- GGC TCT CTG CTC CTC CCTGTT -3′, reverse 5′- GCG GGA TCT CGC TCC TGG AAG -3′; PAR-2: forward 5′- ATT GGT TTG CCC AGT AAT GGT ATG', reverse 5′- AAA AGG ATG GAG CAG TAC ATA TTGC -3′.

## Cell culture and treatment

Cardiac fibroblasts (CFs) were isolated and cultured as outlined by *Golden et al. (2012)*. After 1–3-day-old neonatal SD rats were euthanized, their hearts were cut into small pieces and digested in collagenase type II buffer (1 mg/ml, PH 7.2, Sigma) at 37 °C. This digestion process was repeated until all the tissue blocks were digested. All collected cells were seeded in culture dishes and cultured in DMEM/F-12 (Gibco) containing 5% fetal bovine serum (Gibco). After incubation for 90 min in a humidified atmosphere with 5% $CO_2$ at 37 °C, CFs adhered to the culture dish. The second- and third-generation CFs were used for subsequent experiments. When the cells grew to 70–80% confluence, they were pretreated with RIV (1 ug/ml), FSLLRY (10 uM), or PAR-2 agonist peptide (PAR-2-AP, SLIGKV-NH2; 10 μM, Bachem, Bubendorf, Switzerland) for 1 h, and then exposed to AngII (1 mM; Sigma–Aldrich, Saint Louis, MO, USA) for 24 h. They were then divided into the following groups: control, AngII, RIV+ AngII group, FSLLRY+ AngII group, and the PAR-2-AP+RIV+AngII group.

## Cell counting Kit- 8 assay

The CFs were seeded into 96-well plates at a density of $5\times 10^3$ cells/well. After culturing at 37 °C for 8 h, the cells were treated according to the treatment groups. Cell viability after treatment was detected using the cell counting kit-8 (CCK-8; Bioss, Suzhou, China), according to the manufacturer's instructions. The absorbance was monitored at 450 nm.

## Immunofluorescence (IF)

The CFs were fixed with 4% paraformaldehyde at room temperature for 20 min. Then they were permeabilized with 0.1% Triton X-100 for 15 min and blocked with 5% BSA for 30 min. The CFs were then incubated with the primary antibody α-SMA (1:200, 19245, CST) and Ki67 (1:200, A23722; ABclonal, Woburn, MA, USA) at 4 °C overnight. The next day, after washing with PBS, the CFs were incubated with Alexa Fluor® 488 Labeled Donkey Anti-rabbit IgG (A-21206; Invitrogen, Waltham, MA, USA) for two hours at room temperature, in darkness. The nuclei were stained with 4-6-diamidino-2-phenylindole (DAPI; Invitrogen). The imaging was conducted using a laser confocal microscope (Nikon, Tokyo, Japan). The results were quantified by Image-Pro Plus 6.0 software (Media Cybernetics).

## Statistical analysis

All data is presented as mean $\pm$ standard error of the mean (SEM). The number of biological replicates is specified in the figure/figure legends. Statistical analyses and
graphics were performed using GraphPad Prism 8 (GraphPad Software, Inc.; La Jolla, CA, USA). Nonparametric Mann–Whitney test or one-way ANOVA test were used to compare distributed, and normally continuous variables. A $P$ value $< 0.05$ was considered statistically significant, and individual $p$ values are reported in the figure legends.

## RESULTS

### RIV and PAR-2 antagonist FSLLRY improved cardiac function and hemodynamics in rats with myocardial infarction

Figure 1B shows a representative gross morphology of whole hearts four weeks after MI. In comparison to the sham group, the LAD group exhibited significant areas of ischemic necrosis. However, administering pretreatment with rivaroxaban and FSLLRY resulted in a notable decrease in the ischemic necrosis area when compared to the LAD group. H&E staining revealed that the rats in the LAD group displayed an incomplete structure, disordered arrangement, and focal cytoplasmic vacuolization, which is a hallmark of cell injury, whereas the rats in the RIV and FSLLRY groups showed relatively regular and ordered cardiomyocytes (Fig. 1C). In cardiac catheterization experiments, MI rats showed lower LV end-systolic pressure (LVESP) and lower maximum and minimum dP/dt ($\pm$dP/dT) compared to the rats in the sham group, whereas increased LVESP and higher dP/dt ($\pm$dP/dT) were observed in the RIV and FSLLRY groups (Figs. 2A, 2B). Representative echocardiographic images are presented in Fig. 2C. Echocardiographic examination showed a reduction of ejection fraction (EF) and fractional shortening (FS) in the LAD group (Fig. 2D), suggesting impaired cardiac function, and this impairment was prevented by the RIV treatment. Similar results were observed in the FSLLRY group. These results indicate that RIV and FSLLRY played a beneficial role in improving cardiac hemodynamics and heart function after MI.

### RIV and PAR-2 antagonist FSLLRY attenuated myocardial fibrosis in myocardial infarction rats

Myocardial fibrosis after MI is a feature of adverse left ventricular remodeling. Masson's trichrome staining showed that fibrotic areas were reduced significantly by RIV and PAR-2 antagonist FSLLRY treatment compared to the LAD group (Figs. 1C, 1D). The effect of RIV on the expression of fibrosis markers was further verified by western blot and RT-PCR. Consistent with the results of cardiac histological analyses, the western blot and the RT-PCR results showed that the expression of Collagen I, Collagen III, and α-smooth muscle actin (α-SMA) were upregulated in the LAD group; however, RIV and FSLLRY significantly decreased the expression of these proteins (Figs. 3B–3E). In addition, immunohistochemical staining further showed that α-SMA expression significantly increased in the MI rats, whereas RIV and FSLLRY inhibited the expression of α-SMA in interstitial, but not in peri-vascular after MI (Fig. 3A). Collectively, these data showed that RIV and PAR-2 antagonist FSLLRY could attenuate myocardial fibrosis in MI Rats.

### RIV inhibited Ang II-induced proliferation in cardiac fibroblasts (CFs)

Cardiac fibroblasts (CFs), the primary initiators of the extracellular matrix (ECM), participate in ventricular remodeling through proliferation and migration in response
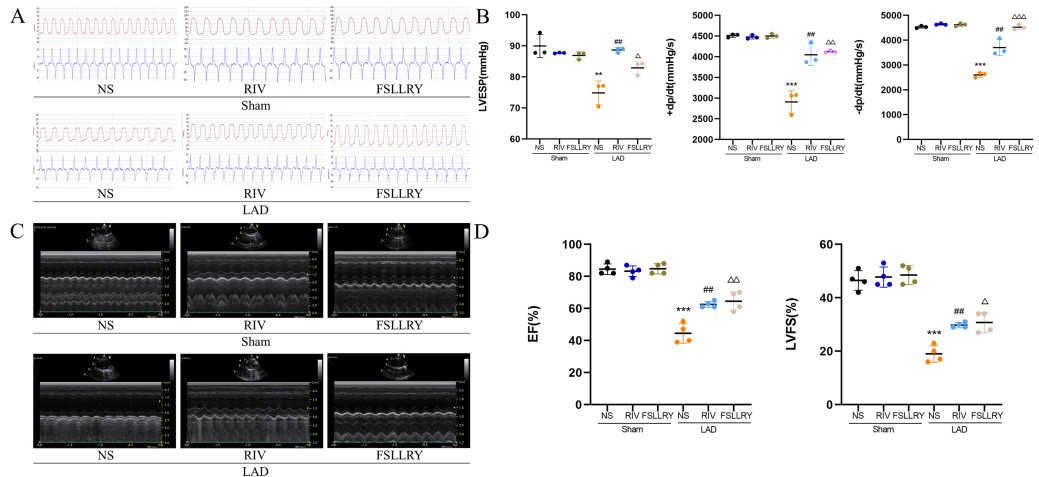

**Figure 2  RIV and FSLLRY improved cardiac function and hemodynamics in MI Rats.** (A) Hemodynamic assessments of rats in each group. (B) Quantitative analysis of hemodynamic parameters. $n = 3$; ***$P < 0.001$, **$P < 0.01$, Sham $vs$ LAD; ##$P < 0.01$, LAD $vs$ RIV; $\triangle\triangle\triangle P < 0.001$, $\triangle\triangle P < 0.01$, $\triangle P < 0.05$, LAD $vs$ FSLLRY. (C) M-mode echocardiography of rats in each group. (D) Quantitative analysis of left ventricular EF and FS determined by echocardiography. $n = 4$; ***$P < 0.001$, Sham $vs$ LAD; ##$P < 0.01$, LAD $vs$ LAD+RIV; $\triangle\triangle P < 0.01$, $\triangle P < 0.05$, LAD $vs$ LAD+FSLLRY.

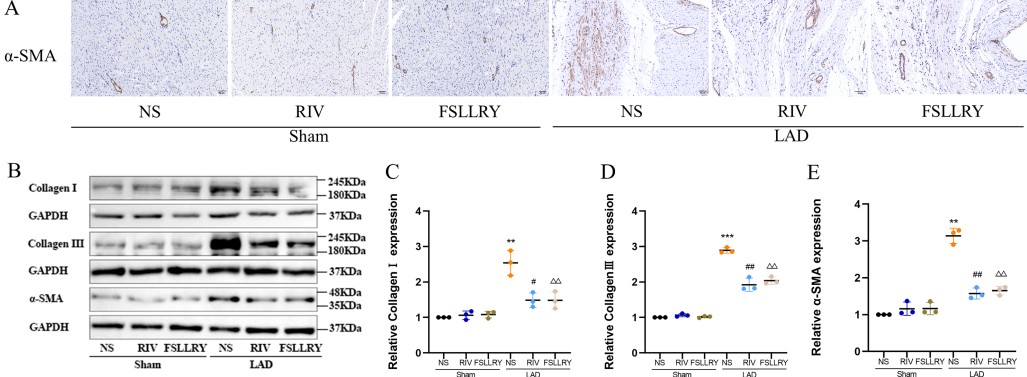

**Figure 3  RIV and FSLLRY attenuated myocardial fibrosis in MI Rats.** (A) Representative images of immunohistochemistry of α-SMA. (B) Representative images of fibrosis-related proteins in the Sham and LAD group treated with RIV and FSLLRY detected by western blotting. $n = 3$; (C–E) Light density assessment of fibrosis-related proteins in the Sham and LAD group treated with RIV and FSLLRY detected by western blotting. $n = 3$; ***$P < 0.001$, **$P < 0.01$, Sham $vs$ LAD; ##$P < 0.01$, #$P < 0.05$, LAD $vs$ LAD+RIV; $\triangle\triangle P < 0.01$, LAD $vs$ LAD+FSLLRY.

to myocardial injury or stress (*Diaz-Araya et al., 2015*). Angiotensin II (Ang II) is a potent pro-fibrotic molecule. Increased serum Ang II levels are seen in patients with cardiovascular disease related to myocardial fibrosis, including acute myocardial infarction, hypertension, and heart failure (*Kim, 2000*). We used Ang II-induced CFs to establish an *in vitro* model of fibrosis. The schematic representation of *in vitro* experiments was showed in Fig. 4A. To detect whether CF proliferation was sensitive to RIV, CFs were induced by Ang II (1

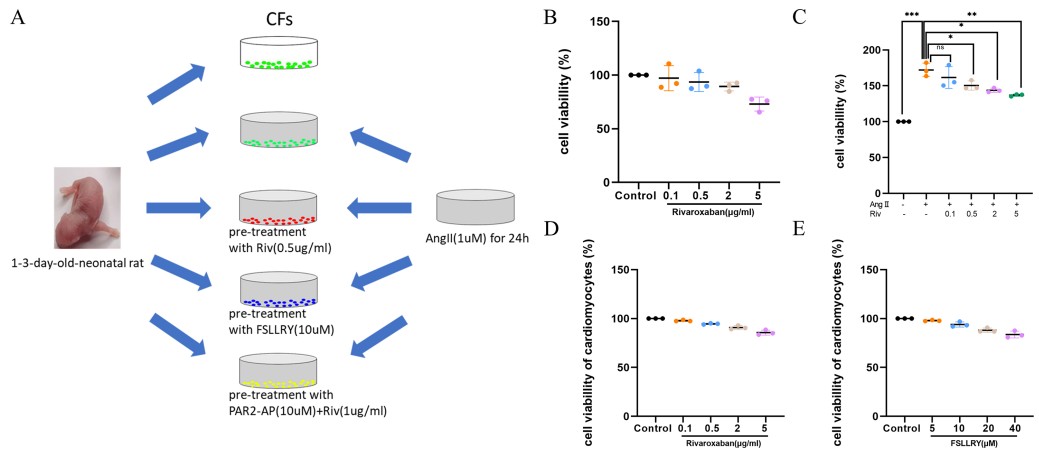

**Figure 4   RIV inhibited Ang II-induced proliferation in CFs.** (A) Schematic representation of *in vitro* experiments. (B) CCK-8 assay detected the CFs viability after induction with different concentrations of RIV. (C) CCK-8 assay detected the cell proliferation of Ang II-induced CFs after pretreatment with different concentrations of RIV. (D) CCK-8 assay detected the cardiomyocytes viability after induction with different concentrations of RIV. (E) CCK-8 assay detected the cardiomyocytes viability after induction with different concentrations of FSLLRY.

μM) and different concentrations (0.1 μg/ml, 0.5 μg/ml, 2 μg/ml, and 5 μg/ml) of RIV were administered. We found CF proliferation significantly increased in the Ang II group compared with the control group, while treatment with RIV reduced the proliferation effects induced by Ang II (Figs. 4B, 4C). We selected RIV at a concentration of 0.5 μg/ml for subsequent experiments due to its higher cell viability and similar inhibitory effect on proliferation as higher concentrations (2 μg/ml, and 5 μg/ml). we also found that both RIV (0.5 μg/ml) and FSLLRY (10uM) had rare effect on the survival of normal cardiomyocytes (Figs. 4D, 4E). These findings suggest that RIV inhibited Ang II-induced CF proliferation.

## RIV attenuated Ang II-induced fibrosis and reduced the expression of PAR-2 in CFs

CFs can be activated to transform into myofibroblasts (MFs), which are characterized by the expression of α-SMA (*Weber & Diez, 2016*; *Leask, 2010*). In addition, these activated CFs can participate in fibrotic responses through proliferation and migration, producing ECM proteins, such as collagen and fibronectin (*Hinz, 2010*; *Weber et al., 2013*). Using immunofluorescence (IF) staining, we found that Ang II stimulation significantly increased the fluorescence intensity of α-SMA and Ki-67 compared with the controls, and this effect was reversed by RIV (Figs. 5A–5D), indicating that RIV could inhibit the Ang II-induced differentiation of CFs into MFs. In CFs stimulated by Ang II, we found that both the mRNA and protein levels of PAR-2 were reduced by RIV, consistent with the downregulation of fibrotic markers (α-SMA, collagen I, and collagen III). Therefore, we explored the possible role of PAR-2 by pretreating CFs with the PAR-2 antagonist (FSLLRY) and PAR-2 agonist peptide (PAR-2-AP). We found that the Ang II-induced upregulation of α-SMA, collagen I and collagen III protein levels was reversed by pre-treatment with FSLLRY. Pretreatment with PAR-2-AP reversed the inhibitory effect of RIV on Ang II-induced fibrosis in CFs

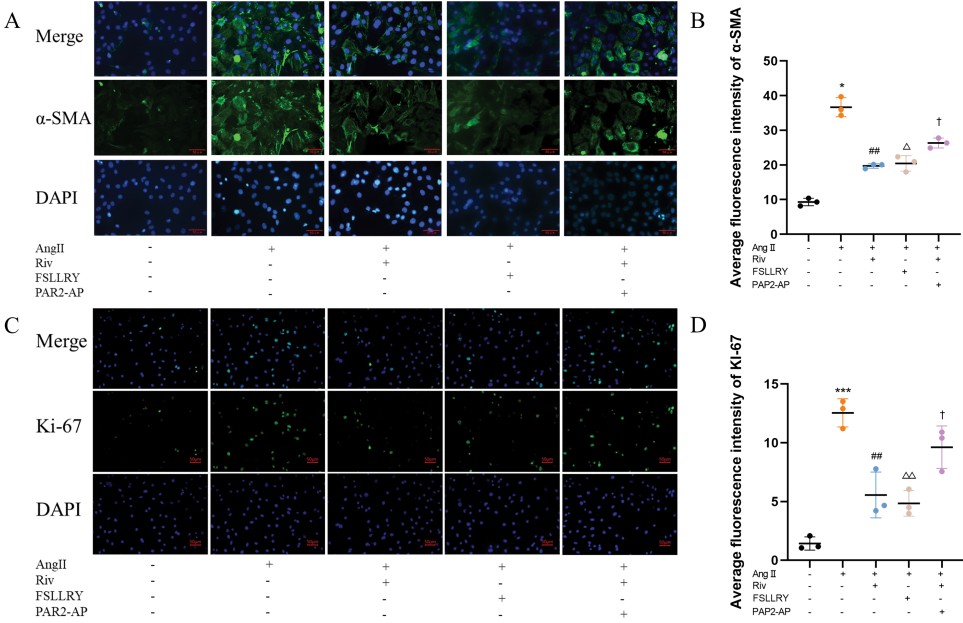

**Figure 5** **RIV inhibited Ang II-induced proliferation in CFs.** (A) The expression of α-SMA was detected by immunofluorescence in the *in vitro* experiments. (B) Quantitative analyses of fluorescent signals for α-SMA. $n = 3$; ** $P < 0.01$, * $P < 0.05$, Control *vs* AngII; ##$P < 0.01$, #$P < 0.05$, AngII *vs* RIV+ AngII; △△$P < 0.01$, △$P < 0.05$, AngII *vs* FSLLRY+AngII. ††$P < 0.01$, †$P < 0.05$, RIV *vs* RIV+ PAR2-AP+AngII. (C) The expression of Ki-67 was detected by immunofluorescence in the *in vitro* experiments. (D) Quantitative analyses of fluorescent signals for Ki-67. $n = 3$; *** $P < 0.01$, ** $P < 0.01$, * $P < 0.05$, Control *vs* AngII; ## $P < 0.01$, # $P < 0.05$, AngII *vs* RIV+ AngII; △△$P < 0.01$, △$P < 0.05$, AngII *vs* FSLLRY+AngII. ††$P < 0.01$, †$P < 0.05$, RIV *vs* RIV+ PAR2-AP+AngII.

(Figs. 6A–6E). IF results also showed that PAR-2-AP reversed the inhibition of α-SMA by RIV (Figs. 5A–5B). These findings revealed that the upregulation of PAR-2 attenuated the inhibitory effect of RIV on Ang II-induced fibrosis in CFs, suggesting that RIV may exert an antifibrotic effect by modulating PAR-2.

## RIV inhibited the TGFβ-related signaling pathway both in myocardial infarction rats and in CFs

TGFβ is a key mediator in the pathogenesis of post-MI remodeling, playing an important role in regulating fibrous tissue deposition and extracellular matrix composition (*Kim, Sheppard & Chapman, 2018*). In our *in vivo* experiments, we used western blot to investigate the expression of the representative proteins in the typical Smads signaling pathway mediated by TGFβ1. We found that the increased TGFβ1 and its downstream Smad2/3 phosphorylation effector induced by MI were both significantly reversed by RIV. The western blot analysis showed that FSLLRY also reversed the expression of TGFβ1 and its pathway proteins (Figs. 7A–7F). These findings indicate that the anti-fibrotic effects of RIV in MI rats are possibly related to its ability to inhibit the TGFβ1 signaling pathway. The above results prompted us to further explore the role of PAR2 in the RIV inhibition TGFβ1 pathway regulation of fibrosis in vitro. We then examined the effects of RIV and FSLLRY on Ang II-induced profibrotic cytokine, TGF-β1, and its main downstream signaling

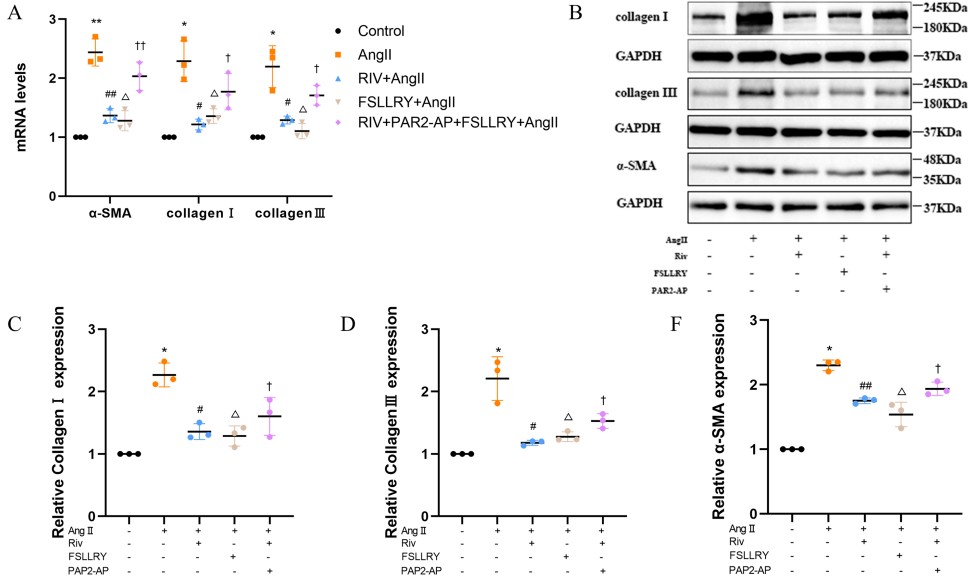

**Figure 6** **RIV attenuated Ang II-induced fibrosis in CFs.** (A) Real-time quantitative PCR analysis of fibrosis-related genes in the vitro experiments. $n = 3$; **$P < 0.01$, *$P < 0.05$, Control *vs* AngII; ##$P < 0.01$, #$P < 0.05$, AngII *vs* RIV+ AngII; △ $\triangle P < 0.01$, $\triangle P < 0.05$, AngII *vs* FSLLRY+AngII. ††$P < 0.01$, †$P < 0.05$, RIV *vs* RIV+ PAR2-AP+AngII. (B) Representative images of fibrosis-related proteins in the vitro experiments detected by western blotting. $n = 3$; (C–F) Light density assessment of fibrosis-related proteins in the vitro experiments. detected by western blotting. $n = 3$; * $P < 0.05$, Control *vs* AngII; ##$P < 0.01$, #$P < 0.05$, AngII *vs* RIV+ AngII; $\triangle P < 0.05$, AngII *vs* FSLLRY+ AngII. †$P < 0.05$, RIV *vs* RIV+ PAR2-AP+AngII.

molecules, p-Smad2 and p-Smad3. We found that, consistent with the findings from our *in vivo* study, both RIV and FSLLRY inhibited TGF-β1 and p-Smad2 and p-Smad3 expressions at the protein level. We pretreated CFs with PAR-2-AP and then administered RIV to explore whether the inhibition of the TGF-β1 pathway by RIV was associated with PAR-2. We found that the inhibiting effect of RIV was partly abolished (Figs. 8A–8E). These findings indicate that the antifibrotic effects of RIV in cardiac remodeling are related to its ability to inhibit the TGFβ signaling pathway, and PAR-2 might be involved in the modulation effects.

## DISCUSSION

The increased understanding of coagulation factors has led to the development of more thrombotic treatment modalities and insights into the non-hemostatic effects of coagulation factors. In this study, we found that RIV attenuated adverse cardiac remodeling and improved cardiac function in the experimental MI rats. Through *in vitro* assays, we found that RIV, *via* PAR-2, blocked the proliferation, differentiation and expression of pro-fibrotic cytokines in CFs and downregulated the TGF-β/smads signaling pathway. These findings indicate that RIV exerted cardioprotective effect against cardiac fibrosis partly through the inhibition of the PAR-2 and TGF-β/smads signaling pathways.

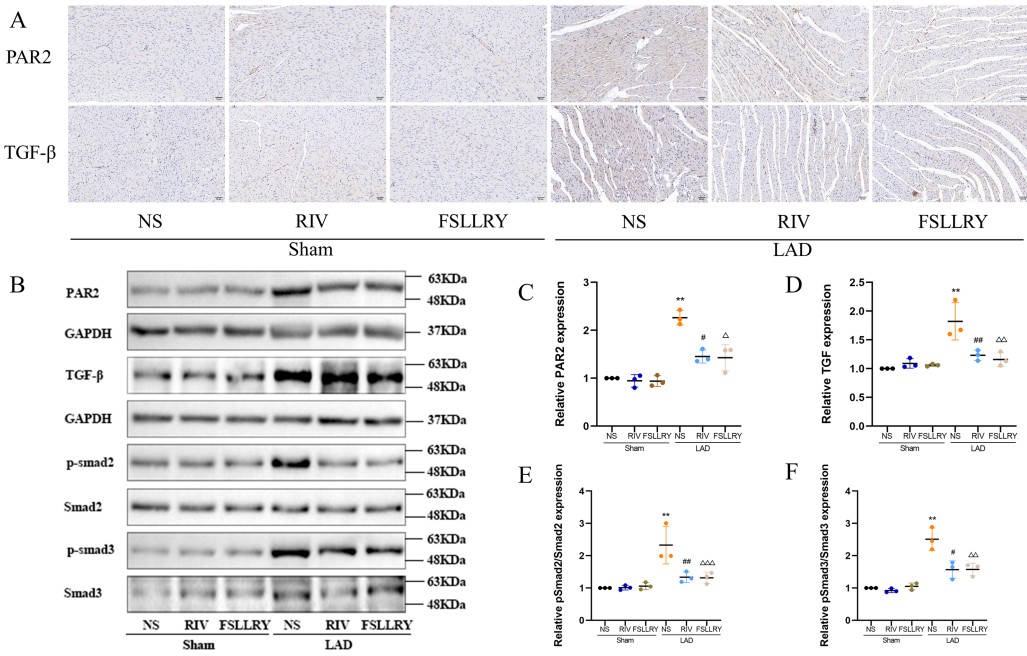

**Figure 7 RIV inhibited TGFβ-related signaling pathways in MI rats.** (A) Representative images of immunohistochemistry of PAR-2 and TGF-β1. (B) Representative images of PAR-2, TGF-β1, p-Smad2, p-Smad3 proteins detected by western blotting. (C–F) Light density assessment of PAR-2, TGF-β1, p-Smad2/Smad2, p-Smad3/Smad3 in the Sham and LAD group treated with RIV and FSLLRY detected by western blotting. $n = 3$; **$P < 0.01$, Sham *vs* LAD; ##$P < 0.01$, #$P < 0.05$, LAD *vs* LAD+RIV; △ △△$P < 0.001$, △△$P < 0.01$, △$P < 0.05$, LAD *vs* LAD+FSLLRY.

Recent studies have shown that RIV is not only therapeutic strategy for the prevention or treatment of thrombosis, but also improves cardiovascular outcomes in patients with coronary artery disease (*Connolly et al., 2018*; *Zhang et al., 2022*). In our MI model, RIV reduced myocardial fibrosis, attenuated cardiac remodeling and the progression of heart failure, as exhibited by reduced myocardial infarct size, ameliorated myocardial pathological damage, restored EF and FS and improved hemodynamic parameters, and markedly reduced cardiac fibrosis and the expression of profibrotic genes and proteins. These data are consistent with previous findings, both in an experimental model of pressure overload (*Kondo et al., 2018*) and in pulmonary hypertension rats (*Imano et al., 2021*), which showed RIV treatment had an antifibrotic effect that was accompanied by cardiac remodeling reversal and reduced fibrotic mediators. Accumulating evidence indicates that the coagulation cascade is involved in adverse cardiac remodeling (*Sechi et al., 2000*; *Zhao & Schooling, 2018*). FXa may exert non-hemostatic effects mediated by PAR-2, with increased expression in response to cardiac injury or stress (*Esmon, 2014*; *Hirano, 2007*). In our study, PAR-2 was increased in the cardiac tissues of MI rats, but this effect was abrogated by RIV treatment. PAR-2 antagonist, FSLLRY, was selected for further elucidation of the potential effect of RIV on PAR-2 activation. The rats treated with FSLLRY exhibited reduced heart remodeling and improved heart function.

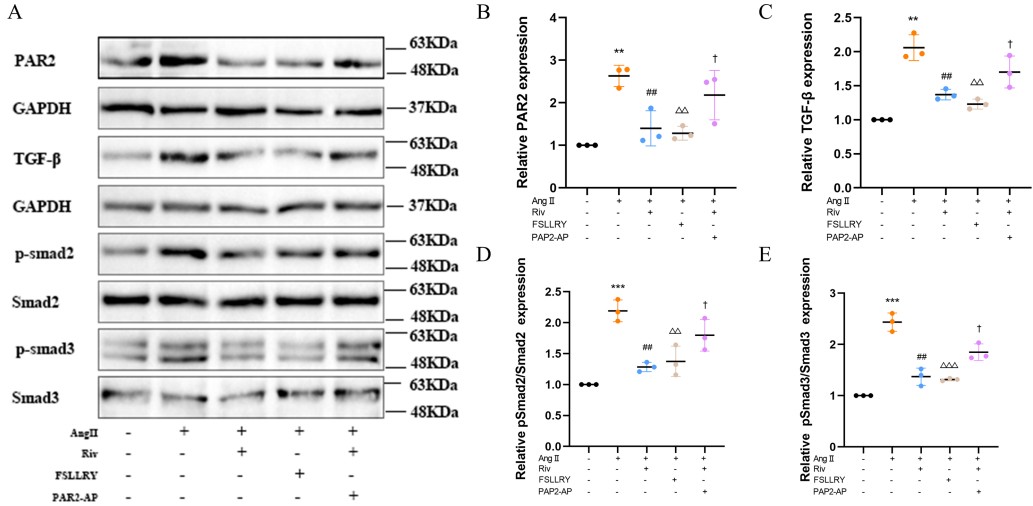

**Figure 8** **RIV inhibited TGFβ-related signaling pathways in CFs.** (A) Representative images of PAR-2, TGF-β1, p-Smad2, p-Smad3 proteins in the *in vitro* experiments detected by western blotting. (B–E) Light density assessment of PAR-2, TGF-β1, p-Smad2/Smad2, p-Smad3/Smad3 in the *in vitro* experiments detected by western blotting. $n = 3$; ***$P < 0.001$, **$P < 0.01$, Control *vs* AngII; ##$P < 0.01$, AngII *vs* RIV+AngII; △△△$P < 0.001$, △△$P < 0.01$, AngII *vs* FSLLRY+AngII. †$P < 0.05$, RIV *vs* RIV+ PAR2-AP+AngII.

Studies have shown that PAR-2 activates multiple intracellular signaling pathways and plays an important role in regulating cardiovascular physiology and pathophysiology (*Ito et al., 2021*; *Ma et al., 2021*). The possible involvement of PAR-2 signaling pathways in cardiac remodeling has also been reported, including one study which found cardiomyocyte-specific PAR-2 overexpression induces left ventricular enlargement and systolic dysfunction (*Antoniak et al., 2013*), and a separate study which found that PAR-2 deficiency attenuates the cardiac systolic dysfunction and fibrosis induced by ischemia/reperfusion injury (*Antoniak et al. 2010*).

To better understand the roles of the PAR-2 signaling pathway in the amelioration of myocardial fibrosis by RIV, we next investigated the role of CFs, which represent the largest interstitial cell population in the heart. Their activity is greatly enhanced after an acute cardiac event or during chronic cardiovascular disease (*Brown et al., 2005*). In the extrinsic coagulation pathway, the tissue factor (TF)/factor VIIa (FVIIa) complex promotes a cascade of proteolytic reactions yielding FXa (*Antoniak, Sparkenbaugh & Pawlinski, 2014*; *Esmon, 2014*). The expression of TF is upregulated in response to Ang II stimulation, and increased TF expression leads to an increase of FXa, which activates the PAR-2 signaling pathway. The PAR-2 signaling pathway activates TF signaling by phosphorylating the TF cytoplasmic structural domain, which can further accelerate the coagulation cascade (*Ahamed & Ruf, 2004*; *Narita et al., 2021*). As expected, in our study, Ang II increased PAR-2 expression in CFs, and RIV or FSLLRY inhibited this increase. RIV and FSLLRY inhibited Ang II-induced CF fibrosis, which was evidenced by the downregulation of the fluorescence intensity expression of α-SMA and the reduction of pro-fibrosis markers. Based on the beneficial effects of PAR-2 inhibitors *in vitro* and *in vivo*, we further found

that PAR-2-AP abrogated the salutary effects of RIV on Ang II-induced fibrosis. Our *in vitro* and *in vivo* results indicate that RIV may inhibit the progression of cardiac fibrosis partly by inhibiting PAR-2 signaling.

It is reported that TGFβ1 plays a central role in mediating myocardial fibroblast activation and excessive extracellular matrix deposition provoked by tissue injury (*Stewart, Thomas & Koff, 2018*). TGFβ1 can be secreted by fibroblasts and acts as a major downstream mediator of angiotensin II action through autocrine signaling (*Rosenkranz, 2004*). Smads proteins are the major effector molecules in the TGF-β signaling pathway (*Tzavlaki & Moustakas, 2020*). Following ligand binding and trans-phosphorylation of TGFβRII, TGFβRI activates Smad2 and Smad3 by phosphorylating specific serine residues in the c-terminal region. Phosphorylated Smad2/3 forms an oligomeric complex with Smad4 and initiates the transcription of genes through transcription into the nucleus (*Massague, 2012*; *Tzavlaki & Moustakas, 2020*). PAR-2 has been reported to promote TGFβ1 signaling by sustaining the protein expression of ALK5 (*Witte et al., 2016*). A relationship between PAR-2 and TGFβ1 has also been found in liver fibrosis and kidney fibrosis. Consistent with these observations (*Chung et al., 2013*; *Knight et al., 2012*; *Sun et al., 2017*), we found that PAR2 antagonist FSLLRY attenuated myocardial fibrosis by inhibiting the TGFβ1 pathway, evidenced by downregulation of TGFβ1 expression and Smad2/3 phosphorylation both in our vivo and *in vitro* studies. Thus, we considered that RIV attenuated myocardial fibrosis due to, at least partially downregulation of PAR2 and TGFβ1 signaling pathway. In order to confirm this, we further found PAR2-AP eliminated part of the AngII-induced the activation of TGFβ1 pathway attenuated by RIV. However, how PAR2 affects the TGFβ1 pathway remain unknown and need to be further explored.

Recent studies have recognized the relationship between coagulation activity and cardiovascular events in humans. The development of drugs that regulate thrombin production or activity provides evidence that the hemostatic system may trigger inflammation, fibrosis, and vascular dysfunction. The findings from our study provide new insights and evidence for the different results seen in recent clinical trials using FXa inhibition in patients with different phenotypes. However, further in-depth studies are needed to confirm the clinical relevance of our observations.

## CONCLUSIONS

Our study demonstrate that RIV exhibits protective roles in MI-induced cardiac remodeling and contractile dysfunction, possibly by inhibiting the activation of PAR-2 and TGF-β1 signaling pathways ameliorating cardiac fibrotic response. The findings provide evidence that RIV may be properly administered in the early stage of heart failure to prevent maladaptive signaling for fibrosis and remodeling, which may provide more theoretical basis for further studies.

**Abbreviations**

| | |
|---|---|
| **PARs** | protease-activated receptors |
| **RIV** | Rivaroxaban |
| **MI** | myocardial infarction |

| | |
|---|---|
| **CFs** | cardiac fibroblasts |
| **LV** | left ventricular |
| **Ang II** | angiotensin II |
| **FXa** | factor Xa |
| **HF** | heart failure |
| **TGFβ1** | transforming growth factor β1 |
| **Ang II** | angiotensin II |
| **LAD** | left coronary artery anterior descending |

## ACKNOWLEDGEMENTS

We thank Jinsha Liu for excellent echocardiography technical support. We also thank Ming Yu for his advice on the isolation and extraction of primary cardiac fibroblasts.

### Funding

This research was funded by the Scientific and Technological Developing Scheme of Ji Lin Province (20210204199YY). The funders had no role in study design, data collection and analysis, decision to publish, or preparation of the manuscript.

### Grant Disclosures

The following grant information was disclosed by the authors:
The Scientific and Technological Developing Scheme of Ji Lin Province: 20210204199YY.

### Competing Interests

The authors declare there are no competing interests.

### Author Contributions

- Qian Zhang conceived and designed the experiments, performed the experiments, analyzed the data, authored or reviewed drafts of the article, and approved the final draft.
- Zhongfan Zhang performed the experiments, authored or reviewed drafts of the article, and approved the final draft.
- Weiwei Chen conceived and designed the experiments, analyzed the data, prepared figures and/or tables, and approved the final draft.
- Haikuo Zheng analyzed the data, prepared figures and/or tables, and approved the final draft.
- Daoyuan Si conceived and designed the experiments, prepared figures and/or tables, authored or reviewed drafts of the article, and approved the final draft.
- Wenqi Zhang conceived and designed the experiments, authored or reviewed drafts of the article, and approved the final draft.
## Ethics

The following information was supplied relating to ethical approvals (i.e., approving body and any reference numbers):

All animal experiments were approved by the Ethics Committee of the First Hospital of Jilin University (protocol code: 0755)

## Data Availability

The raw data is available in the Supplemental Files.

## Supplemental Information

Supplemental information for this article can be found online at http://dx.doi.org/10.7717/peerj.16097#supplemental-information.

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
