# Peer review of "Rivaroxaban, a direct inhibitor of coagulation factor Xa, attenuates adverse cardiac remodeling in rats by regulating the PAR-2 and TGF-β1 signaling pathways"

_PeerJ, doi:10.7717/peerj.16097_

## Round 0.1 · original submission · Major Revisions

I strongly suggest you fully answer all Reviewers' questions and provide all needed changes.

·

Basic reporting

1/ English
To my point of view, the English used is professional. I did not find big mistakes. But, to note, English is not my mother tongue.
I just noticed some points:
- Line 30: a repetition: “by histological staining and histological staining”
- Line 30: “we pretreatment “. It is not correct. Pretreatment is not a verb.
- Line 140: “an-tibodies“. Typing error.
- Line 206: “less significant than that”. Please, rephrase

2/ Literature
In their study, authors aimed to determine the effect of rivaroxaban in the reduction of adverse cardiac remodeling and cardiac function in rat subjected to myocardial infarction. I noticed that, without the in vitro part, a similar study done in wild-type mice, has been recently published (doi:10.1253/circrep.CR-19-0117; Circulation Report, 2020). This publication is surprisingly not in the reference of the article we review. It could be nice to compare the results obtained in the two studies mainly regarding the different time course. Another study, done in PAR-2-/- mice model gave interesting results that can be also used here for a comparison/discussion (doi:10.1016/j.thromres.2018.05.015; Thrombosis research 2018). This study is also not referenced.

3/ Background/context
The background is correct but not sufficient to understand their point. Indeed, it miss the discussion of the effects known of rivaroxaban in myocardial infarction recovery. At the end of their introduction authors noted “Details concerning the mechanism by which of rivaroxaban attenuates adverse cardiac remodeling have not been fully elucidated”. It is true but some recent publications give some possible pathways/ mechanisms. For instance, Nobuhiro Nakanishi and colleagues demonstrated that rivaroxaban protects against cardiac dysfunction in myocardiac infarction model by, possibly, reducing levels of PAR-1 and PAR-2 levels (they checked at mRNA level) and proinflammatory cytokines in the infarcted area. This pathway is not mentioned in the background. But, it is an important point for this study. To the best of my knowledge, the research article we are reviewing is the first using a rat model of myocardiac infarction to study the effect of rivaroxaban. It is a plus-value for this article. Nevertheless, several pathways have been elucidated in mice and should be write on the background.
4/ Article shape
The structure of the article is highly professional. Raw data are present and explained

Experimental design

1/ The article is well on the scope of the journal.

2/ The question is a good one and is interesting. The weakness of their hypothesis is coming from the fact that a previous study did more or less the same in mice. Nevertheless, to the best of my knowledge, it is the first study like this done in rats. Furthermore, authors searched also pathways in vitro.

3/ Methods:
- Microscopes used to take pictures of the histological slides and the immunofluorescence are mentioned but not the one used for the immunohistochemistry slides. Why?

- Statistics: Authors indicated: “T-test was used for comparison between two independent samples”. Which one?

- Graphes: To allow readers to see the dispersion of the data, it is better to use individual values for each graphe.

- Line 96, authors noted: “All animals were treated humanely”. What means for them “humanely”? They already mentioned to have an ethic approval. Authors can develop this idea. Or remove it. As they want. An approvement by an ethical commity is sufficient. Of course, animals have not to suffer but authors fill the ARRIVE form.

- Methods are generally well described, except for the Immunohistochemical analysis of myocardial (IHC) part. Authors have to describe this part with more details.

Validity of the findings

I have different and large comments/questions regarding the results. Mainly, looking the western blots.

1/ Figure 3, panel A. The images are too over exposed. Of course, we can discriminate the staining. I did not see positive staining on sham figure. Nevertheless, the main problem of this overexposition is the observation of the staining observed for the rivaroxaban and the FSSLRY. It also miss a negative control with only the secondary antibody. I did not find it in supplementary data.

2/ Figure 6, panel A, it also miss a negative control with only secondary antibody.

3/ Western-blots
In all the panels of figure 3, figure 5, figure 6, and figure 7 presenting immunoblottings, only one GAPDH line is shown for all the gels. It is trendy now in scientific publications, but, to my point of view it is a mistake. Indeed, it is impossible for the reviewers and the readers to be sure of the western-blots quantification.
Authors gave copies of the original membranes. It is mentioned that all the western blot has been done in triplicate noted A, B, and C by the authors. I will use the same names.

- Figure 3:
 Collagen 1: the bands presented in the main data for the collagen 1 is coming from triplicate A
 Collagen 3: the bands presented in the main data for the collagen 3 is coming from triplicate C
 Alpha-SMA: the bands presented in the main data for the alpha-SMA is coming from triplicate B
 GAPDH: the bands presented in the main data for the GAPDH is coming from triplicate B. It means that the GAPDH bands is corresponding only to the alpha-SMA bands presented.

Question 1: Are the 3 proteins quantified on the same western-blots? In case of a positive answer, due to the close molecular weight of collagen 1 and 3, authors probably used glycine or other compound to remove primary antibody. This kind of technique is often used but more or less never mentioned on the material and methods part. If the authors used this technique they have to mentioned it. Besides, there is a possibility that glycine or other compound used to remove primary antibody on a membrane before hybridization with another one can heterogeneously remove a part of the protein. Did the authors used GAPDH after glycine if they used?

Question 2: In supplementary data, authors noted: “Statistical data comparisons were obtained from 3 independent replicate western blot experiments with duplicates of collagen I, collagen III, α-SMA and GAPDH”. Why the duplicates are not present on the membranes and not shown? Where are the duplicates?

- Figure 5:
 Collagen 1: I did not find the blot where the band of collagen I are coming from. I also noted a problem of molecular weight on membrane A (raw data)
 Collagen 3: the bands presented in the main data for the collagen 3 seem to come from triplicate A but the exposition is different. The bands presented in main data have not the same exposition.
 Alpha-SMA: the bands presented in the main data for the alpha-SMA is coming from triplicate A
 GAPDH: the bands presented in the main data for the GAPDH is coming from triplicate A. It means that the GAPDH bands is corresponding only to the alpha-SMA bands presented.

- Figures 5 and 6: PAR-2 bands have a problem of homogeneity.

For all the blots presented in main data, it could be nice show the molecular weights ( a range: 63-48 Kda or 245-180 Kda) and not only the molecular weight of the protein of interest. It will be more rigorous for your data presentation.

Regarding the conclusion, it is highly correct. Not ultra-interpolation of their data.

Additional comments

Dear authors,

Experimentally it is mandatory to have controls, at least a negative one for the immunohistochemistry.
For the western blot, it could be nice to present the duplicates but also GAPDH band for each independant protein. It is widely used now to mix the GAPDH bands. But is is a mistake. And, if we want to have a correct scientific litterature, it could be nice to avoid this kind of mistake.

To my point of view, without these revisions (in balance between major and minor revisions), your paper cannot be published.

·

Basic reporting

The manuscript contains numerous grammatical errors that may detract from the overall quality of the work. It is recommended that the author consider utilizing a professional editing service to ensure that the manuscript is free from errors and is presented in the best possible manner.

Experimental design

Overall, this study is well-designed. However, it is recommended that the potential toxicity of Rivaroxaban and FSLLRY on normal cardiomyocytes be addressed in order to provide a more complete understanding of the effects of these compounds.

Validity of the findings

The results presented in this study have addressed the study question.

Additional comments

Minor concerns should be addressed.

1. In this manuscript, the authors demonstrate that Rivaroxaban and FSLLRY can inhibit cardiac fibroblasts (CFs). It would be beneficial to include a toxicity test of Rivaroxaban and FSLLRY on normal cardiomyocytes in Figure 1, in order to provide a more comprehensive analysis of the effects of these compounds.
2. Figure 3A should include the quantification of a-SMA intensity or percentage.
3. Figure 4D should include Ki67 staining, and the quantification of a-SMA intensity or percentage should be included.
4. The authors should clarify how intracellular TGF-β affects the Smads signaling pathway. Should cell autocrine TGF-β affect the Smads signaling pathway, right? Accurate descriptions should be provided in the manuscript.
5. The figure legend for Figure 6 should be corrected to match the figures. Specifically, Figures 6B and 6C should be exchanged.
6. The reason for the different bands in p-Smad3 in Figure 6B and Figure 7A should be addressed in the manuscript.

·

Basic reporting

The manuscript was professionally written. It was clear and easy to follow for the most part. There were some grammatical errors in the manuscript (for example, line 30; it should be ‘we pretreated’ not ‘we pretreatment’, line 74; ‘PAR-2 works synergistically’ not ‘PAR-2 synergistically’, etc.) that needs to be proofread and addressed to make this manuscript publication ready.

Sufficient background was provided on the subject. Background on Rivaroxaban and MI model was also elucidated in the introduction. However, a few key publications were not referenced in the introduction which were relevant to the manuscript. These publications include Nakanishi, Nobuhiro, et al. Circulation reports 2.3 (2020): 158-166; Mitsuishi, Rintaro, et al. Journal of the American College of Cardiology 69.11S (2017): 2033-2033 and Imano, Hideki, et al. Journal of pharmacological sciences 137.3 (2018): 274-282.

Article structure conformed to the PeerJ guidelines. All sections were defined appropriately and followed the prescribed format. One suggestion to the authors would be to include an abbreviation table. The readers will greatly benefit from one. A labeling issue was observed with Figure 5. Although the title mentioned that Rivaroxaban reduced the expression of PAR2 in CFs, no data related to the expression of PAR2 was shown in this figure. Moreover, multiple replicates of the IHC images which were mentioned in the methods should be included in the supplementary figures.

The publication is self-contained as this constitutes as a minimum unit of publication although the rationale for the redundancy of the publication can be questioned which is elucidated in a later section.

Experimental design

The original scope and aim of the research is well defined but it fails to identify the knowledge gap and how this research is fulfilling it. Some of the publications referenced in the previous section has covered the effect of Rivaroxaban in MI model mice and the underlying mechanism of PAR2 and TGFβ in the MI models both in vivo and in vitro. So a major part of the design would be to identify the gap and how this publication is going to move the understanding of this field forward which they have failed to define.

The investigation was performed at a high ethical standard. The techniques are well defined and the experiments were performed at a high technical standard.

Methods were well defined except for the light density assessment which was not described in the Methods section. Moreover, the use of Image J to assess the light density assessment was elaborated in the methods section as well.

Validity of the findings

As mentioned in the previous sections, this study is derivative of some earlier works on Rivaroxaban in MI as shown in this publication without referencing some of these studies. Rationale for replication of these findings were not clearly elucidated in the introduction or study design. How this study adds value to the existing knowledge was also not elaborated efficiently.

The data was presented in the publication in a way that can be easily comprehensible, with clear description of the experiment and the findings of them. Only instance where the description was problematic was in Figure 5 where PAR2 was mentioned but the relevant data was not shown until later in the figure lists. Also, in the in vivo study, Rivaroxaban and FSLLRY arms must be present for both LAD and sham studies. The effect of these drugs in the sham would be controls and may have served as background to the actual data. Furthermore, CFs being able to differentiate to MFs using Ang II treatment was not referenced and no data was shown to suggest the differentiation process was completed. The differentiation was not described in detail in the paper which would make it difficult to replicate this result. Lastly, Collagen mentioned in the manuscript was spelled as Col1agen instead of Collagen which has to be corrected.

Most of the conclusions were based on the findings of the paper except for the claims made in Line 293 and 303. No assessment of the cardiac function was done on the treated animals or the cells. So the claims in these lines about the treatment method improving cardiac functions sounds farfetched since no specific experiment was done to assess that.

Additional comments

One area of major improvement for the manuscript would be to build on the already existing knowledge of the studies and delve deeper into the mechanism of action of the drug itself as well as the understanding of the MI pathways to make this study more comprehensive and fill the gaps in knowledge of this particular area of study.

---

## Round 0.2 · Major Revisions

I suggest carefully taking into consideration the Referee's comments on the negative controls. I also recommend reporting the statistical tests used for the analysis in detail.

·

Basic reporting

English is clear and professional.
Same note regarding the structure.
I have a comment regarding the figures.

Experimental design

Experiment design is well defined. Authors presented results of sham experiments. These results are usefull to interpret the data.
Nevertheless, it miss negative controls for all experiments. It miss rigour.

Validity of the findings

It exists controled experiments but not negative constrols.
Nothing is mentioned about precise statistical tests used.
To my point of view it is better if graphes are done with plots.

·

Basic reporting

no comment

Experimental design

no comment

Validity of the findings

no comment

Additional comments

I acknowledge and appreciate the authors' sincere efforts to bring this manuscript.The authors have done a very good job of addressing the previous concerns.

---

## Round 0.3 · accepted · Accept

The Review found that all changes were made appropriately.

·

Basic reporting

Ok

Experimental design

Ok

Validity of the findings

Ok

Additional comments

After the second reviewing, authors gave pictures of negative controls and better explained statistics.

For me all is fine now.